# Comparative Biochemical and Transcriptome Analyses Reveal Potential Candidate Genes Related to Pericarp Browning in Red Rice

Gileung Lee [1,2], Jae Wan Park [1], Jisu Park [1], Ah-Reum Han [1], Min Jeong Hong [1], Yeong Deuk Jo [3], Jin-Baek Kim [1], Sang Hoon Kim [1] and Hong-Il Choi [1,*]

1   Advanced Radiation Technology Institute, Korea Atomic Energy Research Institute, Jeongeup 56212, Republic of Korea
2   Crop Breeding Division, National Institute of Crop Science, Rural Development Administration, Wanju 55365, Republic of Korea
3   Department of Horticultural Science, Chungnam National University, Daejeon 34134, Republic of Korea
*   Correspondence: hichoi@kaeri.re.kr

**Abstract:** Proanthocyanidins (PAs) are major phytochemicals in rice, and accumulate abundantly in red pericarp (RP) rice. Consumers and rice breeders are increasingly showing interest in PAs because of their beneficial health effects; however, PA biosynthesis in rice is not well-understood. Therefore, to gain insights into this process, we performed comparative transcriptome analysis of grains harvested at 14 days after flowering (DAF; i.e., the stage at which active PA biosynthesis occurs without pericarp color change) and 28 DAF (the stage of late seed development with pericarp color change) from RP and white pericarp rice. In RP rice at 14 DAF, the expression levels of six structural genes (*OsCHS*, *OsF3H*, *OsF3′H*, *OsDFR*, *OsANS*, and *OsLAR*), one modification gene (*OsUGT*), and one transport gene (*OsGSTU34*) were significantly upregulated, along with the activation of *Rc*, the key regulator of PA accumulation in the pericarp. Functional enrichment analysis of 56 differentially expressed genes specifically upregulated in RP rice at 28 DAF revealed the presence of three laccase genes known to cause the browning reaction through oxidation. These results expand our understanding of PA biosynthesis in rice, and provide a genetic basis that will lead to further studies on the genes and underlying molecular mechanisms associated with this process.

**Keywords:** *Oryza sativa*; seed coat browning; proanthocyanidin; RNA sequencing; laccase

## 1. Introduction

In addition to the common white color, rice has various other pericarp colors, including black, brown, and red, depending on the type and combination of phytopigments accumulated in the pericarp. Although the white pericarp (WP) is predominant in cultivated rice varieties due to artificial selection according to human preference during domestication, colored rice has steadily been cultivated and used as food, in medicine, and in religious ceremonies in Asian counties [1]. In addition to attractiveness of its colorful appearance, colored rice has a higher nutritional value than WP rice because it contains various bioactive compounds, including phenolic acids, carotenoids, γ-oryzanol, and vitamin E [2]. Moreover, anthocyanins and proanthocyanidins (PAs) abundantly accumulate in colored rice, and are known to have beneficial effects on human health, e.g., antioxidant, anticancer, and antiinflammatory activities, lowering cholesterol and blood glucose, and antiadhesive mechanisms against viruses and bacteria [3,4]. Given its nutritional value, colored rice is increasingly gaining interest among health-conscious consumers and rice breeders who seek to increase the nutritiousness of rice.

PAs are major phytochemicals in rice that accumulate mainly in the pericarp of red rice. These are also known as condensed tannins because they are polymeric phenolic

compounds produced by the polymerization of flavan-3-ol units, such as catechin and epicatechin [5]. PAs are synthesized through a series of enzymatic steps from the phenyl-propanoid pathway to the flavonoid pathway, and they share the upstream biosynthesis pathway of anthocyanin, in which phenylalanine is converted into leucoanthocyanidins through stepwise catalytic reactions [6]. The leucoanthocyanidins (uncolored) are converted to 2R,3S-flavan-3-ols (uncolored) by leucoanthocyanidin reductase, or to anthocyanidins (colored) by anthocyanin synthase, and then to 2R,3S-flavan-3-ols (uncolored) by antho-cyanidin reductase; both flavan-3-ols are the precursor units of PAs [3,4]. The biosynthetic structural genes of PAs are under sophisticated control by multiple regulatory genes that encode transcription factors with MYB domains, basic helix–loop–helix (bHLH) domains, or WD-40 repeats. These three types of regulatory genes are known to act together to form the MYB–bHLH–WD40 (MBW) complex that regulates the biosynthesis of PAs [6].

The precursors of PAs synthesized via the biosynthesis pathway are initially colorless, but are subsequently converted to reddish-brown PAs through polymerization and oxi-dation reactions [7]. It is thought that an oxidation reaction due to polyphenol oxidase or peroxidase is the main cause of PA browning [3]. In Arabidopsis, seed coat browning is observed during the seed desiccation period in the late stages of seed development, and it has been reported that *TRANSPARENT TESTA 10/AtLAC15*, which encodes laccase-like polyphenol oxidase, may participate in the oxidative polymerization of PAs and regulate seed coat browning [8,9]. However, despite decades of research, the exact mechanisms un-derlying PA polymerization and oxidation, and the enzymes involved in these mechanisms, are yet to be clarified.

Because of the domestication process, WP has become the dominant phenotype of rice, resulting in relatively insufficient research on PA biosynthesis. *Rc* and *Rd* have been identi-fied as key genes related to the red pericarp (RP) phenotype caused by PA accumulation in rice. *Rc* encodes a bHLH transcription factor, which is a regulatory gene that controls the expression of PA biosynthetic structural genes and is a representative rice domestication gene responsible for the determinant of PA accumulation in the pericarp [10]. *Rd* encodes dihydroflavonol 4-reductase, one of the structural genes involved in PA biosynthesis path-way, and regulates color deepening of the pericarp through an epistatic interaction with *Rc* [11]. Most research related to pigmented rice has been focused on black rice (containing large amounts of anthocyanins) rather than red rice; therefore, there is a lack of data on PA biosynthesis. In the present study, we performed RNA sequencing-based transcrip-tome analysis of two japonica genotypes with RPs and WPs, respectively, to improve our understanding of PA biosynthesis in rice pericarps.

## 2. Materials and Methods

### 2.1. Plant Materials

RBRC_OS_RP1, a japonica breeding line with an RP and *RcRc* genotype, and Dongan, a japonica cultivar with a WP and *rcrc* genotype, were used in this study. Plant materials were grown under normal cultivation conditions in an experimental paddy field at the Advanced Radiation Technology Institute, Jeongeup, Korea. Rice grain samples were collected at 7, 14, 21, 28, and 35 days after flowering (DAF), as well as at full maturity, for 4-dimethyl aminocinnamaldehyde (DMACA) staining and quantitative analysis of total phenolic and tannin content. The samples collected at 14 and 28 DAF were used for transcriptome analysis.

### 2.2. 4-Dimethylaminocinnamaldehyde Staining

Grain samples harvested at different stages of maturity were immediately soaked in absolute ethanol for decolorization. After 2 h of decolorization, the samples were stained with 1% (*w/v*) DMACA (Sigma-Aldrich, St. Louis, MO, USA) in a mixture of ethanol (Merck, Darmstadt, Germany) and 6-M hydrochloric acid (1:1, *v/v*; Sigma-Aldrich) for 2 h. Stained samples were rinsed with several changes of 70% ethanol.

### 2.3. Quantification of Total Phenolic and Tannin Content

Freeze-dried grain samples (1 g of each sample) were individually extracted using 10-mL methanol and sonication for 1 h. Each methanol extract was filtered through Whatman No. 4 filter paper for analysis. The standards, namely ferulic acid and tannin acid, were weighed accurately and dissolved in methanol to 1 mg/mL. The stock solutions were then diluted to produce a series of standard solutions at seven different concentrations (6.25–500.00 μg/mL) to assess total phenolic content (TPC) and total tannin content (TTC). To measure the TPC in each sample solution, a methanol sample extract (0.1 mL) was mixed with 1 mL Folin–Ciocalteu reagent, neutralized using 0.8 mL saturated sodium carbonate (75 g/L), and incubated at room temperature for 2 h. Absorbance was measured at 765 nm using a UV–Vis spectrophotometer (Evolution 260 Bio; Thermo Fisher Scientific Inc., Waltham, MA, USA). A standard calibration curve was prepared with a ferulic acid equivalent, and the TPC was calculated as milligrams of ferulic acid equivalent per gram of sample. To measure the TTC, 1 mL of sample extract was mixed with 1 mL distilled water, 1 mL 95% ethanol, 1 mL 5% sodium carbonate, and 0.5 mL 1-N Folin–Ciocalteu reagent. After 1 h, the absorption was measured at 725 nm using the UV–Vis spectrophotometer. A standard calibration curve was prepared with a tannic acid equivalent, and the TTC was calculated as milligrams of tannic acid equivalent per gram of sample.

### 2.4. RNA Extraction and Sequencing

Total RNA was extracted from grain samples (collected at 14 and 28 DAF) using an RNeasy Plant Mini Kit (Qiagen, Hilden, Germany) according to the manufacturer's instructions. The quantity of RNA was measured using a NanoDrop ND-1000 spectrophotometer (NanoDrop Technologies, Wilmington, DE, USA), and equal amounts of total RNA from three biological replications were pooled before library construction. RNA quality was checked on a 2100 Bioanalyzer (Agilent, Santa Clara, CA, USA), and RNA Integrity Number values were confirmed as > 9 for all samples. The cDNA library for RNA sequencing was constructed using a TruSeq RNA Library Preparation Kit (Illumina, San Diego, CA, USA) according to the manufacturer's instructions. The cDNA libraries were subsequently sequenced in a single lane of an Illumina HiSeqX platform to generate 150-bp paired-end reads. The raw reads generated via sequencing were subjected to preprocessing steps. Adaptor sequences were trimmed using cutadapt, and low-quality sequences (phred score < 20) and short-length reads (read length ≤ 25 bp) were removed using DynamicTrim and LengthSort in the SolexaQA package. After preprocessing, cleaned reads were mapped to the rice reference genome (IRGSP-1.0) using HISAT2. Read count, i.e., the number of reads per gene, was calculated, and normalization was conducted to estimate gene expression levels using HTSeq [12] and DESeq [12].

### 2.5. Identification of Differentially Expressed Genes and Enrichment Analysis

Differentially expressed genes (DEGs), according to comparisons of target transcriptome datasets, were identified based on read count, binomial tests, and fold change (FC) in gene expression. Genes with a false discovery rate (FDR)-adjusted $p$-value of <0.01 and |log2 FC| of ≥1.5 were considered DEGs. The DEGs were subjected to Gene Ontology (GO) enrichment analysis using G:Profiler version e104_eg51_p15_39838c3 with the g:SCS method (FDR < 0.05) for multiple testing correction of $p$-values [12]. The list of GO terms was summarized by eliminating redundant terms using the REVIGO tool [13] with the SimRel semantic similarity measure and a cutoff value of C = 0.5. Additionally, Kyoto Encyclopedia of Genes and Genome (KEGG) pathway enrichment was performed using KOBAS 3.0 [14], and KEGG pathways with a FDR of <0.05 were considered significantly enriched pathways.

### 2.6. Validation of Gene Expression via Quantitative Reverse Transcription Polymerase Chain Reaction

For some DEGs thought to be related to PA biosynthesis in RP rice, RNA sequencing-based gene expression was validated using quantitative reverse transcription PCR (qRT-

PCR). RT-qPCR was performed using ExcelTaq 2X Fast Q-PCR Master Mix (SMOBIO Technology, Hsinchu, Taiwan) on a CFX96 Real-Time System (BioRad, Hercules, CA, USA) following the manufacturer's instructions. For each target gene, expression data were generated using three biological replicates with three technical replicates per biological replicate. Expression levels were normalized using the ΔCt method, based on the expression level of *OsACT1*, as an internal control. Information on primers is provided in Supplementary Table S1.

### 2.7. Statical Analysis of Experimental Data

Experiments were carried out at least in triplicate. Significant differences were estimated using Student's *t*-test ($p < 0.05$) in Microsoft Excel 2019.

## 3. Results

### 3.1. Pericarp Color Change and Proanthocyanidin Accumulation during Grain Development

To estimate the characteristics of pericarp color according to grain development, grains were harvested at six developmental stages (7, 14, 22, 28, and 35 DAF, and at full maturity). In the early stages of development, the pericarp of WP rice was pale green; however, the green color gradually faded and finally turned white as grain development progressed (Figure 1A). Up to 21 DAF, the pericarp of RP rice was also pale green and did not differ from WP rice in this respect; however, the pericarp color changed to reddish-brown at 28 DAF and deepened gradually until full maturity (Figure 1B). Grains of RP and WP rice harvested at each stage were stained with DMACA, which reacts with PA precursors and PAs. Although the seeds of WP rice were not stained with DMACA at any stage, those of RP were stained blue at all stages from 7 DAF (Figure 1C,D). Additionally, TPC and TTC, including PAs as constituents, were measured to estimate the temporal change in PA content in grain samples (Figure 1E). Both the TPC and TTC of RP rice rapidly increased up to 14 DAF, with the highest content recorded at 4.87 mg FA/g and 4.24 mg TA/g, respectively, after which both gradually decreased as grain-filling progressed. Thus, the biosynthesis of PAs proceeds actively in the early seed developmental stages, but the accumulation of PAs in the pericarp does not directly lead to color change.

### 3.2. RNA Sequencing-Based Transcriptome Profiling and Detection of Differentially Expressed Genes

RNA sequencing was performed using grain samples of RP and WP rice from 14 DAF (i.e., the active PA biosynthesis stage) and 28 DAF (i.e., the pericarp browning stage) to investigate the pigmentation of red rice. In total, 95.8 million raw reads generated from four libraries (RP14, RP28, WP14, and WP28) were filtered through preprocessing, resulting in 91.3 million clean reads constituting 12.2 Gb of sequence data. The clean reads were aligned to the reference genome (Nipponbare: IRGSP-1.0) with a mapping rate of 86.9–89.2% (Supplementary Table S2). For comparison of gene expression according to the difference in pericarp color, RP and WP transcriptome datasets at the same stage were determined as contrast groups (RP14 vs. WP14 and RP28 vs. WP28). In total, 401 and 908 DEGs were detected in the 14 (RP14 vs. WP14) and 28 DAF (RP28 vs. WP28) contrast groups, respectively (Figure 2A). The 14 DAF DEG set included 150 and 251 upregulated and downregulated DEGs, respectively (Figure 2B and Supplementary Table S3), whereas the 28 DAF DEG set included 342 and 566 upregulated and downregulated DEGs (Figure 2B and Supplementary Table S4).

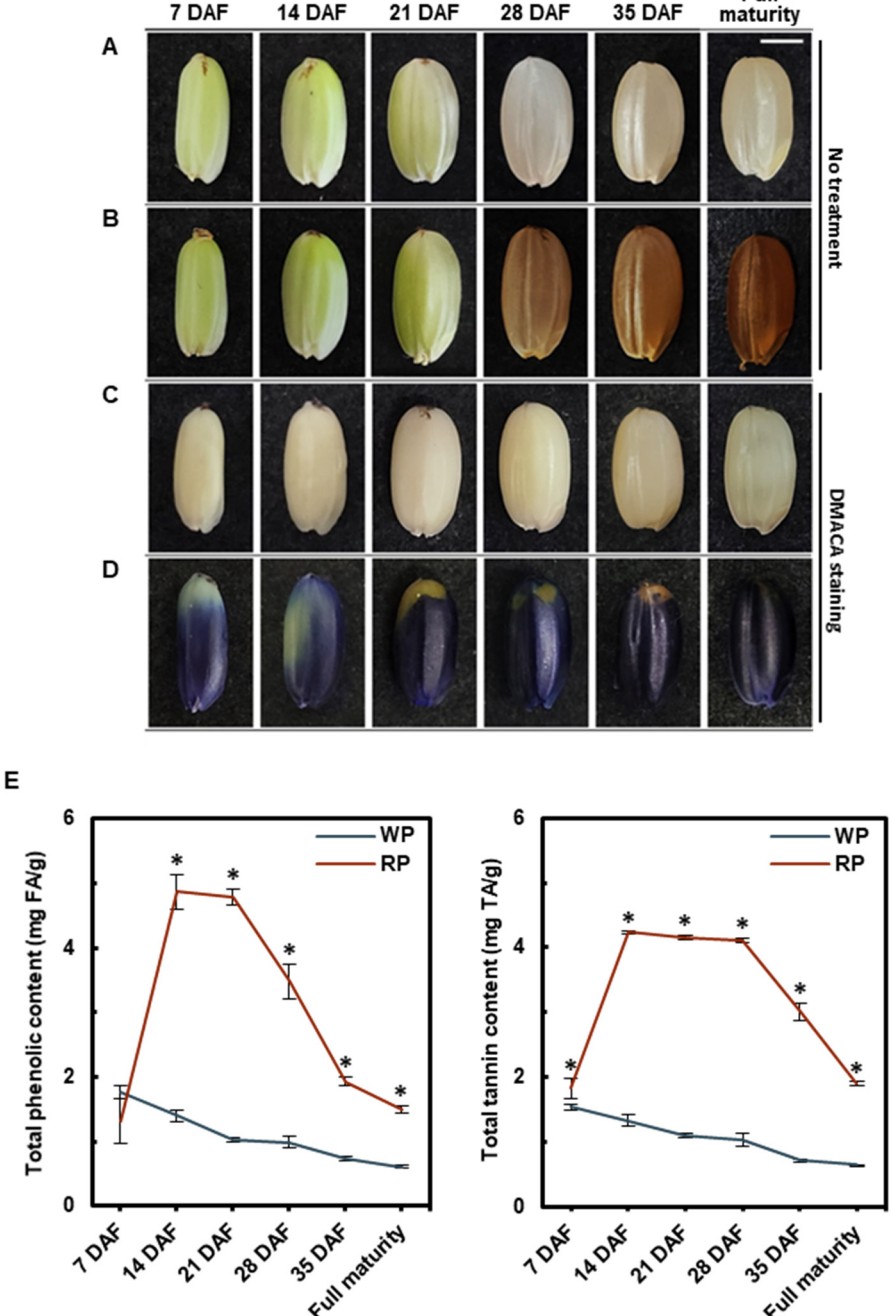

**Figure 1.** Comparison of grain characteristics between red pericarp (RP) and white pericarp (WP) rice. Pericarp color of WP (**A**) and RP (**B**) rice according to seed development. Pericarp color changes in WP (**C**) and RP (**D**) rice after 4-dimethyl aminocinnamaldehyde (DMACA) staining. Scale bar: 2 mm. (**E**) Quantitative analysis of total phenolic content and total tannin content in seeds of WP and RP rice during seed development. Error bars represent the standard deviation of three replicates. Asterisks denote significant differences according to Student's *t*-test (*p* < 0.05). FA, ferulic acid; TA, tannin acid.

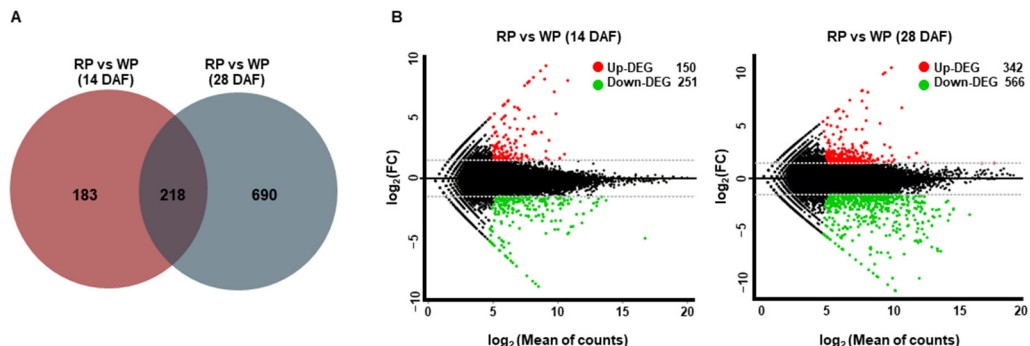

**Figure 2.** Differentially expressed genes (DEGs) between the seeds of red pericarp (RP) and white pericarp (WP) rice. Venn diagram (**A**) and MA plot (**B**) showing the number of DEGs between RP and WP rice at different seed developmental stages. Red and green dots represent significantly upregulated and downregulated genes, respectively; black dots represent genes with no significant difference.

*3.3. Gene Ontology and Kyoto Encyclopedia of Genes and Genome Pathway Enrichment Analysis*

To classify the function of DEGs, the upregulated and downregulated DEG sets were divided and subjected to GO enrichment analysis. The 14 DAF upregulated DEGs were significantly enriched in the GO Biological Process (GO-BP) terms—flavonoid biosynthetic process (GO:0009813), DNA strand elongation (GO:0022616), and mitotic DNA replication initiation (GO:1902975) as well as the GO Molecular Function (GO-MF) term—DNA replication origin binding (GO:0003688) (Figure 3A). The 28 DAF upregulated DEGs were significantly enriched in the GO-MF terms—oxidoreductase activity (GO:0016491), hydrolase activity_hydrolyzing O-glycosyl compounds (GO:0004553), and hydrolase activity_acting on glycosyl bonds (GO:0016798), as well as the GO Cellular Component (GO-CC) terms—cell periphery (GO:0071944) and intrinsic component of membrane (GO:0031224) (Figure 3B). In total, 24 GO terms, including 17 GO-BP, 5 GO-MF, and 2 GO-CC terms, were enriched in the 14 DAF downregulated DEG set, whereas 31 GO terms, including 14 GO-BP, 15 GO-MF, and 2 GO-CC terms, were enriched in the 28 DAF downregulated DEG set (Figure 3 and Supplementary Table S5). Several GO terms (13 GO-BP, 5 GO-MF, and 1 GO-CC terms) were enriched in both downregulated DEG sets, most of which were related to stress and protein binding and folding. KEGG pathway analysis showed that the 14 DAF upregulated DEGs were enriched in flavonoid biosynthesis (dosa00941) and DNA replication pathways (dosa03030), whereas the 28 DAF upregulated DEGs were enriched in diterpenoid biosynthesis pathways (dosa00904). The 14 and 28 DAF downregulated DEGs were enriched in protein processing in the endoplasmic reticulum pathway (dosa04141), and most of the genes enriched in this pathway encoded heat shock proteins (Table 1 and Supplementary Table S6).

**Table 1.** Kyoto Encyclopedia of Genes and Genome pathway enrichment analysis of differentially expressed genes between red and white pericarp rice.

| Stage | DEG Set | KEGG ID | Pathway Name | Count | Expected Count | FDR |
|---|---|---|---|---|---|---|
| 14 DAF | Upregulated | dosa00941 | Flavonoid biosynthesis | 6 | 41 | $7.12 \times 10^{-7}$ |
| | Upregulated | dosa03030 | DNA replication | 4 | 57 | $1.09 \times 10^{-3}$ |
| | Downregulated | dosa04141 | Protein processing in endoplasmic reticulum | 20 | 210 | $3.91 \times 10^{-15}$ |
| 28 DAF | Upregulated | dosa00904 | Diterpenoid biosynthesis | 4 | 41 | $4.25 \times 10^{-2}$ |
| | Downregulated | dosa04141 | Protein processing in endoplasmic reticulum | 27 | 210 | $2.95 \times 10^{-14}$ |
| | Downregulated | dosa04626 | Plant-pathogen interaction | 10 | 192 | $2.75 \times 10^{-2}$ |

Count: the number of DEGs belonging to a pathway; expected count: the total number of genes in a pathway.

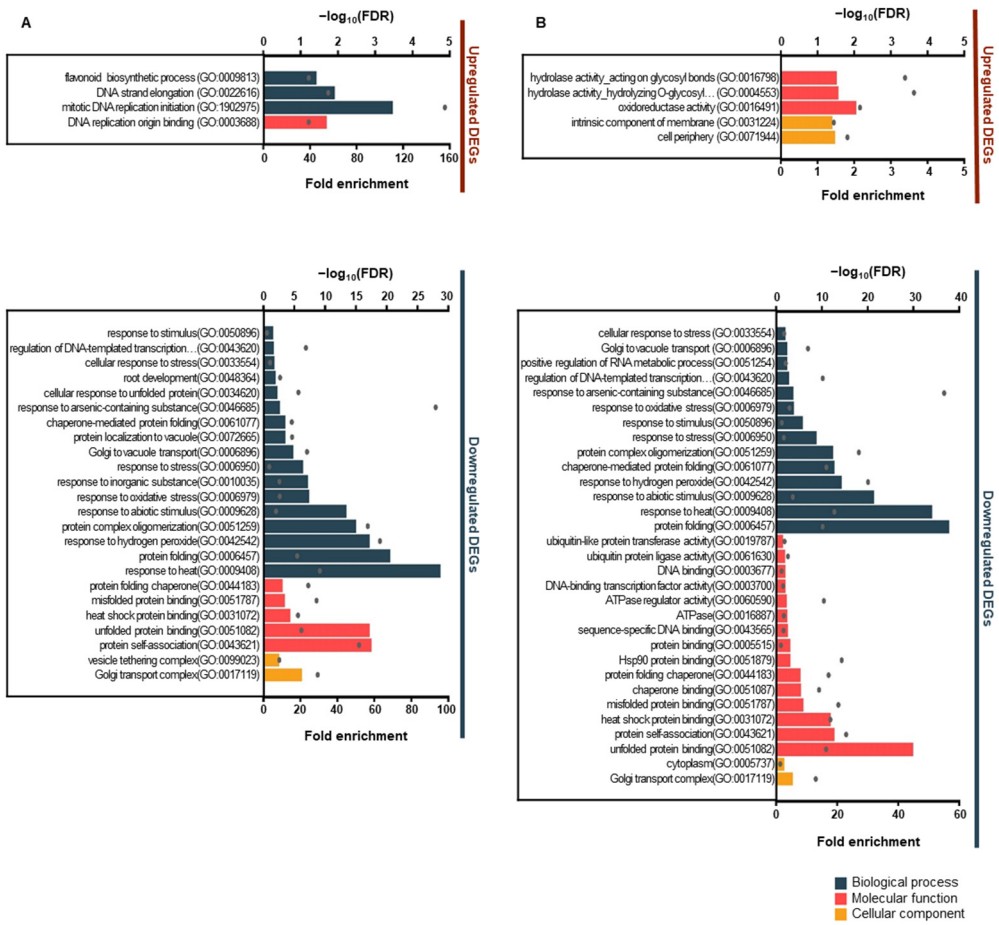

**Figure 3.** Gene Ontology (GO) enrichment analysis of differentially expressed genes (DEGs). Enriched GO terms of upregulated and downregulated DEGs between red pericarp (RP) and white pericarp (WP) rice at 14 days after flowering (DAF) (**A**) and 28 DAF (**B**). Bar charts represent the $-\log_{10}$ false discovery rate (FDR) of enriched GO terms with an FDR of <0.05, and the gray dots denote fold enrichment.

### 3.4. Expression Profile of Rice PA Biosynthesis-Related Genes

GO and KEGG enrichment analyses revealed that genes in the flavonoid biosynthesis pathway, which is a major pathway in PA biosynthesis, were enriched in the 14 DAF upregulated DEGs. To identify genes that affect in RP rice, expression profiling of 23 genes known to be involved in PA biosynthesis was performed, and the expression levels in the transcriptome datasets and FC values between RP and WP rice genes were visualized as a heatmap (Figure 4 and Table S7). Three genes belonging to the phenylpropanoid pathway, *OsPAL*, *OsC4H*, and *Os4CL*, were highly expressed in both RP and WP rice without differential expression. In the flavonoid pathway, six structural genes, including *OsCHS*, *OsF3H*, *OsF3'H*, *OsDFR*, *OsANS*, and *OsLAR*, were differentially upregulated in RP rice at 14 DAF. In particular, the expression of *OsF3H* and *OsDFR* markedly increased in RP rice compared with WP rice at both 14 and 28 DAF. Among five regulatory genes, *Kala3*, known as the regulatory gene in anthocyanin biosynthesis, was differentially upregulated in RP28, and *Rc*, known as a key regulatory gene in PA biosynthesis, was differentially upregulated in both RP14 and RP28. Unlike *Kala3*, *Rc* was rarely expressed in WP rice, but was highly expressed in RP rice, similar to the expression patterns of *OsF3H*, *OsDFR*, *OsANS*, *OsLAR*, and *OsGSTU34*. The expression levels of DEGs were decreased at 28 DAF compared with 14 DAF, consistent with the phytochemical content results according to seed development. Thus, PA accumulation in the pericarp of RP rice is apparently determined by the expression of structural genes in the flavonoid pathway, and *Rc* may play a pivotal role in regulating the expression of these DEGs.

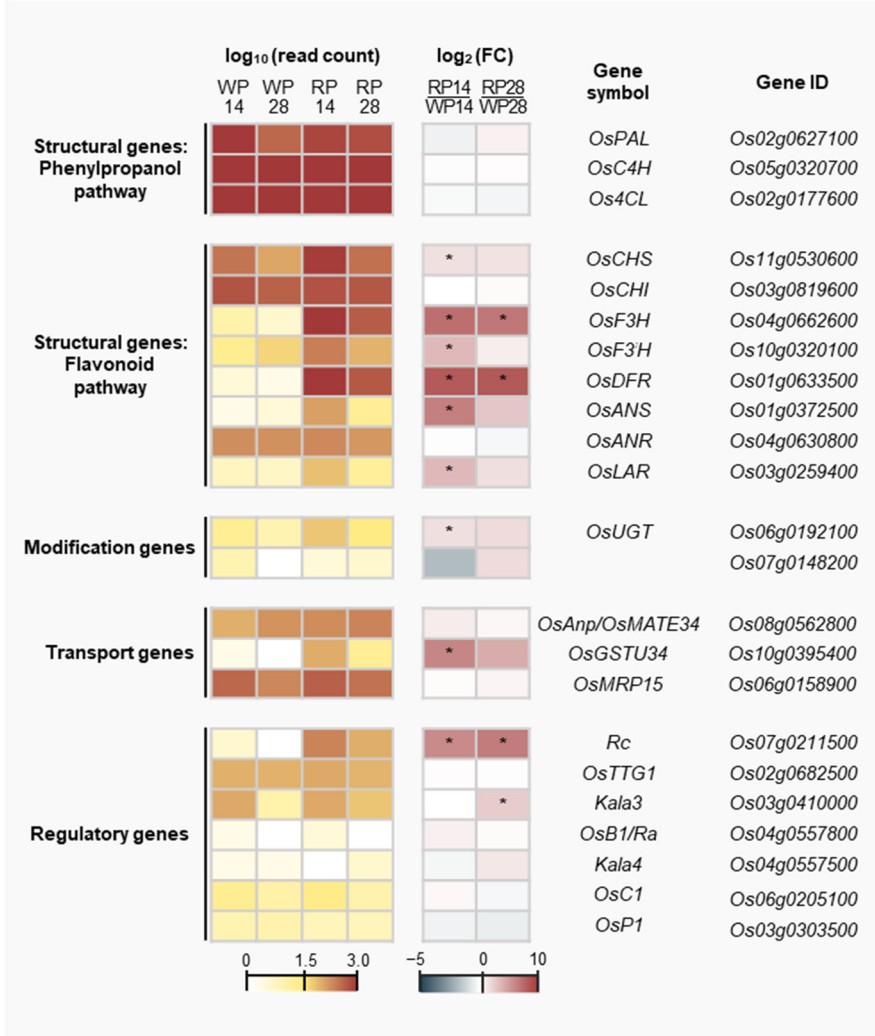

**Figure 4.** Heatmap of selected PA biosynthesis-related genes in two different seed developmental stages. Asterisks indicate significant differences between the expression levels of red pericarp (RP) and white pericarp (WP) genes.

### 3.5. Potential Candidate Genes Related to Pericarp Browning

To identify candidate genes involved in pericarp browning, we screened 56 DEGs specifically upregulated in RP28 based on the following criteria: intersection among RP28 vs. RP14 upregulated DEGs, RP28 vs. WP14 upregulated DEGs, and RP28 vs. WP28 upregulated DEGs (Figure 5A and Supplementary Table S8). GO enrichment analysis of these 56 DEGs revealed five significantly enriched GO terms: lignin catabolic process (GO:0046274) and phototropism (GO:0009638) in BP, and three oxidoreductase activity-related terms in MF (GO:0052716, GO:0016679, and GO:0016722) (Figure 5B). Interestingly, four of the five enriched GO terms were commonly derived from three DEGs, *Os11g0641500*, *Os12g0258700*, and *Os12g0257600*, all of which encoded laccase protein, a type of polyphenol oxidase (Figure 5B). Considering previous reports that colorless PAs turn reddish-brown through a browning reaction caused by an oxidation reaction, these are promising candidate genes that may play a major role in the pericarp browning of red rice.

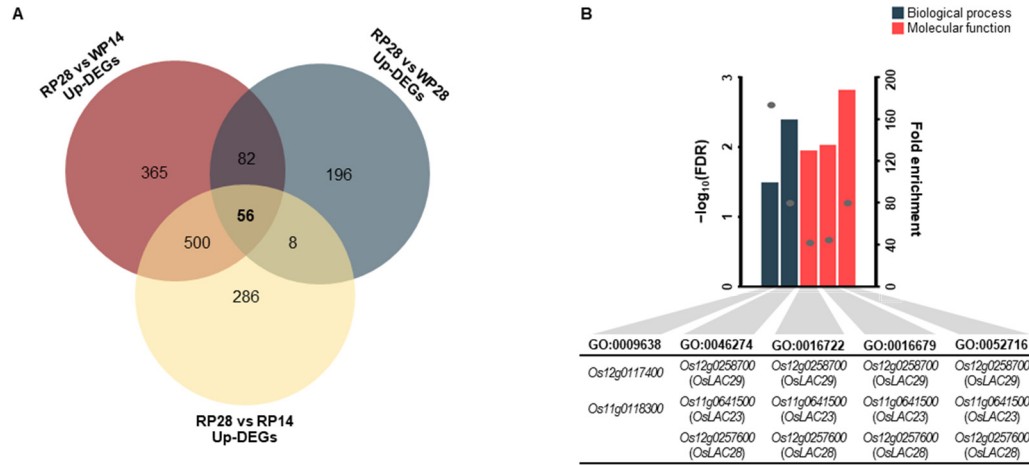

**Figure 5.** Analysis of RP28-specific upregulated differentially expressed genes (DEGs). Venn diagram (**A**) of the upregulated DEGs in RP28 vs. RP14, RP28 vs. WP14, and RP28 vs. WP28. GO enrichment analysis (**B**) of RP28-specific upregulated genes. GO:0009638, phototropism; GO:0046274, lignin catabolic process; GO:0016722, oxidoreductase activity, acting on metal ions; GO:0016679, oxidoreductase activity, acting on diphenols and related substances as donors; GO:0052716, hydroquinone:oxygen oxidoreductase activity. Bar charts represent the $-\log_{10}$ false discovery rate (FDR) of enriched GO terms with an FDR of <0.05, and the gray dots denote fold enrichment.

### 3.6. Quantitative Reverse Transcription-Polymerase Chain Reaction Validation of RNA Sequencing-Based Expression Data

To verify the results of RNA sequencing analysis, qRT-PCR was performed on several DEGs expected to affect PA biosynthesis. Eleven DEGs, including six flavonoid pathway genes (*OsCHS*, *OsF3H*, *OsF3′H*, *OsDFR*, *OsANS*, and *OsLAR*), one transport gene (*OsGSTU34*), one regulatory gene (*Rc*), and three potential candidate genes related to the pericarp browning process (*OsLAC23*, *OsLAC28*, and *OsLAC29*) were selected for RT-qPCR analysis. All genes related to PA accumulation showed significantly different expression levels in RP and WP rice at 14 DAF, and the expression levels were decreased at 28 DAF. In contrast, candidate genes related to the browning process were highly expressed in the late stage of seed development. The expression pattern of each gene obtained using qRT-PCR at 14 and 28 DAF was highly consistent with that obtained in RNA sequencing (Figure 6).

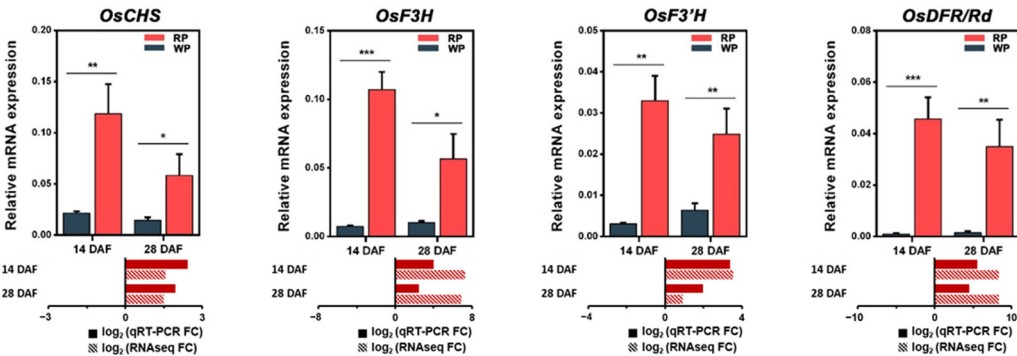

**Figure 6.** *Cont.*

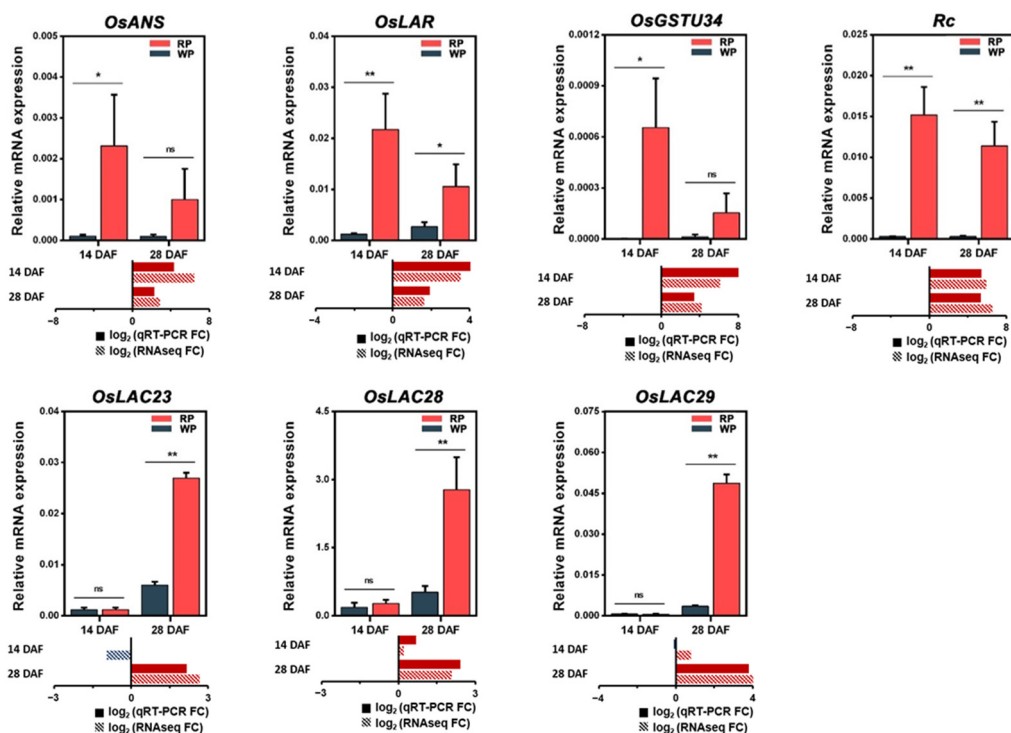

**Figure 6.** Quantitative reverse transcription-polymerase chain reaction (qRT-PCR) validation of several differentially expressed genes (DEGs) related to proanthocyanidin biosynthesis. Vertical bar charts indicate relative gene expression normalized using the internal control gene *OsACT1*. Horizontal bar charts show the fold change values between red pericarp (RP) and white pericarp (WP) rice genes according to RNA sequencing and qRT-PCR. Error bars represent the standard deviations of three biological replicates. Asterisks denote significant differences according to Student's *t*-test (* $p < 0.05$, ** $p < 0.01$, *** $p < 0.001$).

## 4. Discussion

PAs are major phytochemicals and representative bioactive substances that provide plants with protection from external predation, as well as providing nutritional benefits to forage crops and edible crops [3]. However, the PA biosynthesis process in rice is not well-known compared with that in other plants. Therefore, we conducted comparative transcriptome analysis in RP and WP rice to expand our understanding of PA biosynthesis and to identify candidate genes related to the pericarp browning process.

Unlike anthocyanins, which are reddish to purple, PAs are colorless at the time of synthesis and require a browning process for conversion into colored PAs [15]. Hence, a temporal gap is often observed from PA biosynthesis in tissue to the appearance of colored PAs in that tissue. In Arabidopsis, uncolored PAs are detected at an early stage of seed development, i.e., the two-cell embryo stage in the seed coat, but the color change in the seed coat is observed around 10 days after pollination, concomitant with seed desiccation [8]. In the present study, the accumulation of PAs in the pericarp of RP rice was detected using DMACA staining from 7 DAF, but the pericarp color remained pale green, as in WP rice, until 21 DAF. Subsequently, the pericarp color of RP rice gradually turned reddish-brown (Figure 1). Additionally, the TPC and TTC in RP seeds rapidly increased from 7 to 14 DAF, but gradually decreased thereafter according to the progress of grain filling. Because PAs belong to a class of tannin and polymeric phenolic compounds, and are known to be positively correlated with TPC [15,16]. PA content may also increase steeply until 14 DAF based on the results of TPC and TTC analysis in RP rice. Based on these results, we infer that PAs are actively synthesized in the early stage of seed development, and that the browning reaction occurs in the late stage of seed development. Accordingly,

transcriptome analysis of the seeds of RP and WP rice was performed at 14 (i.e., the PA biosynthesis stage) and 28 DAF (i.e., the PA browning stage).

The flavonoid pathway constitutes a central pathway in the PA biosynthesis pathway [3]. We found that flavonoid biosynthesis terms (GO:0009813 and dosa00941) were commonly enriched in 14 DAF upregulated DEGs (Figure 3A and Table 1), indicating that structural genes in the flavonoid biosynthesis pathway might play important roles in PA biosynthesis. Six structural genes, *OsCHS*, *OsF3H*, *OsF3′H*, *OsDFR*, *OsANS*, and *OsLAR*, were upregulated in RP14 and showed a decreasing expression pattern at 28 DAF (Figure 4), implying that PA biosynthesis actively occurs in the early stages of seed development, but is reduced in late stages and is mainly regulated by flavonoid biosynthesis pathway genes. Our expression data is fairly consistent with transcription profile analyses previously reported in other red rice accessions, particularly for *OsCHS*, *OsF3H*, *OsF3′H*, *OsDFR*, and *OsLAR* [17,18]. Among the flavonoid pathway structural genes, *OsCHS*, *OsF3H*, *OsF3′H*, *OsDFR*, and *OsLAR* were commonly upregulated in RP rice across all studies. Thus, these commonly upregulated structural genes are likely essential for the expression of the RP trait and are regulated by a highly conserved regulatory system.

Glucosyltransferases are enzymes that decorate anthocyanidins, resulting in modified anthocyanins with diverse colors and stabilities [19]. In PA biosynthesis, epicatechin, the building block of PAs, is modified to epicatechin 3′-O-glucoside by glucosyltransferase, resulting in transportation to the vacuole via the multidrug and toxic compound extrusion (MATE)-mediated transport process [20]. In rice, *OsUGT* (*Os06g0192100*) and *Os07g0148200* have been reported as orthologues of Arabidopsis *ANTHOCYANINLESS1*, which is involved in the glycosylation of anthocyanin, but glucosyltransferase with specificity for epicatechin has not yet been reported [21]. In our transcriptome data, *Os07g0148200* was rarely expressed in both RP and WP rice, whereas *OsUGT* was upregulated in RP14 relative to WP14, and showed an expression pattern similar to that of *Rc* (Figure 4). Therefore, *OsUGT* could be involved in the PA modification process under the regulation of *Rc* in the rice pericarp.

After the PA biosynthetic process occurs at the endoplasmic reticulum, the units of PAs and PAs are exclusively transported into vacuoles [22]. To date, several genes encoding proteins, such as MATE transporter, glutathione-S-transferase (GST), and H+-ATPase, have been found to participate in PA transportation and vacuole sequestration [23]. In the transport gene category, *OsGSTU34*, which encodes GST, was significantly upregulated in RP14, similar to the upregulation of structural genes regulated by *Rc* (Figure 4). Previously, functional characterization of *OsGSTU34* revealed that the gene was involved in anthocyanin accumulation in the leaves of black rice, but it is not known whether *OsGSTU34* is associated with PA transportation in the rice pericarp [24]. However, given that several GST genes, such as *TT19*, *VviGST3*, *VviGsT4*, and *AcGST1*, play roles as transporters of Pas, as well as anthocyanins in various plant species [25], it can be inferred that *OsGSTU34* may also be involved in PA transportation.

Despite sharing a biosynthesis pathway and intermediates with various flavonoids, such as anthocyanins, PAs are tissue specifically accumulated through regulatory mechanisms controlled by several types of transcription factors and their complexes. In Arabidopsis, the TT2 (MYB)/TT8 (bHLH)/TTG1 (WD40) complex is the main MBW complex that regulates PA accumulation in the seed coat, and three additional complexes with partially overlapping transcription regulatory functions are known to participate in this regulation in a tissue-specific manner [26]. In rice, *Rc*, which encodes a bHLH protein, has been previously reported as a major regulatory gene in PA biosynthesis [10]. However, additional regulatory genes, either those acting alone or through interaction with *Rc*, have not been reported to date. *Rc* is undeniably the central regulatory gene of PA accumulation, given that the WP and RP colors of rice germplasm are accurately distinguished by the *Rc* allele type [27] and that the expression of the RP trait can be determined by genetically manipulating *Rc* [28]. In the temporal expression profiling of *Rc* conducted previously, it was found to play a regulatory role during early seed development, and the

functional allele displayed a higher level of expression than that of the defective allele via a self-upregulation mechanism [17]. This is consistent with our transcriptome data, in which *Rc* was rarely expressed in WP rice and was more actively expressed at RP14 than at RP28 (Figure 4).

The browning process is typically caused by the oxidation of phenolic compounds, and the oxidation reaction is known to be catalyzed by polyphenol oxidases [29]. To identify candidate genes related to the browning process, we screened genes specifically upregulated in RP28, which was the only sample with an RP of the four samples used in our transcriptome analysis. GO enrichment analysis of 56 DEGs specific to RP28 revealed the presence of three laccase genes, which encode a type of polyphenol oxidase. Laccases are members of the multicopper oxidase enzyme family, which catalyze the oxidation of phenolic substrates [30]. In plants, laccase functions in various processes, including lignification in cell walls, wound healing, responses to stress, seed yield determination, and seed coat color determination. In Arabidopsis, *TT10/AtLAC15* was identified as the gene that controls oxidative browning of the seed coat, and its function is well-characterized [9]. Additionally, *BnTT10*, *ADE/LAC*, and *LAC14-4* have also been reported as browning-related genes in rapeseed, litchi, and longan, respectively [31–33]. In rice, *OsLAC10* and *OsLAC13* are reported to play roles in the Cu stress response and lignin biosynthesis and the regulation of seed setting rate, respectively [34], but a laccase gene that affects pericarp color browning has not yet been reported. Therefore, our findings provide important new insights into the association between laccase and the browning reaction in rice.

## 5. Conclusions

Comparative biochemical and transcriptomic analyses of RP and WP rice were performed; accordingly, we revealed different expression levels of PA biosynthesis-related genes in RP and WP rice and identified potential candidate genes that may be responsible for PA biosynthesis. In particular, *OsLAC23*, *OsLAC28*, and *OsLAC29* were specifically upregulated in RP28 undergoing pericarp browning, implying that they are promising candidate genes for the browning reaction in rice. Overall, these findings broaden our understanding of PA biosynthesis and pigmentation in color rice, and will help facilitate in-depth research into the identity and function of the genes involved in this process, which has not yet been fully characterized.

**Supplementary Materials:** The following supporting information can be downloaded at: https://www.mdpi.com/article/10.3390/agriculture13010183/s1, Table S1: Primer information for quantitative reverse transcription-polymerase chain reaction, Table S2: Summary of RNA sequencing data and mapping statistics, Table S3: Differentially expressed genes between red and white pericarp rice at 14 days after flowering, Table S4: Differentially expressed genes between red and white pericarp rice at 28 days after flowering, Table S5: Gene Ontology enrichment analysis results for upregulated and downregulated differentially expressed genes between red and white pericarp rice, Table S6: Kyoto Encyclopedia of Genes and Genome enrichment analysis results for upregulated and downregulated differentially expressed genes between red and white pericarp rice, Table S7: Information on rice PA biosynthesis-related genes, Table S8: Intersecting genes among the upregulated differentially expressed genes of RP28 vs. RP14, RP28 vs. WP14, and RP28 vs. WP28.

**Author Contributions:** Conceptualization, H.-I.C. and G.L.; resources, H.-I.C.; investigation, G.L., J.W.P., J.P., A.-R.H. and M.J.H.; writing—original draft preparation, G.L.; writing—review and editing, A.-R.H., M.J.H., Y.D.J. and H.-I.C.; funding acquisition, J.-B.K. and S.H.K.; supervision, H.-I.C. All authors have read and agreed to the published version of the manuscript.

**Funding:** This research was supported by the the research program of Korea Atomic Energy Research Institute (Project No. 523310-23).

**Institutional Review Board Statement:** Not applicable.

**Informed Consent Statement:** Not applicable.

**Data Availability Statement:** Data is contained within the article or supplementary material.

**Conflicts of Interest:** The authors declare no conflict of interest.

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
