# Peer review of "Comparative Biochemical and Transcriptome Analyses Reveal Potential Candidate Genes Related to Pericarp Browning in Red Rice"

_agriculture, doi:10.3390/agriculture13010183_

Round 1
Reviewer 1 Report
Author performed comparative transcriptome analysis of grains harvested at 14 days after flowering and 28 DAF from RP and white pericarp rice. The results help to understanding of PA biosynthesis in rice.
1.Transcriptome sequencing was performed on the samples of two time periods of the two cultivars the RBRC_OS_RP1 and Dongan.It should be noted how many repetitions were made
2. To verify the results of RNA sequencing analysis, qRT-PCR was performed. Eleven DEGs, including six flavonoid path way genes, one transport gene, one regulatory gene, and three potential candidate genes related to the pericarp browning process (OsLAC23, OsLAC28, and OsLAC29) were selected for RT-qPCR analysis. Why are the three candidate genes of LAC gene family. This needs to be explained in detail and analyzed for reasons.
Author Response
Reviewer 1
Author performed comparative transcriptome analysis of grains harvested at 14 days after flowering and 28 DAF from RP and white pericarp rice. The results help to understanding of PA biosynthesis in rice.
Response: We appreciate the reviewer’s time and effort to review our manuscript.
1.Transcriptome sequencing was performed on the samples of two time periods of the two cultivars the RBRC_OS_RP1 and Dongan. It should be noted how many repetitions were made
Response: We have already described sample preparation for RNA sequencing in Materials and Methods (lines 127-128). We prepared three biological replicates (three panicles from three plant individuals) harvested at each developmental stage. After extracting RNA independently from grains of each triplicates, these were pooled in equal amounts to prepare one RNA sample for sequencing.
2. To verify the results of RNA sequencing analysis, qRT-PCR was performed. Eleven DEGs, including six flavonoid path way genes, one transport gene, one regulatory gene, and three potential candidate genes related to the pericarp browning process (OsLAC23, OsLAC28, and OsLAC29) were selected for RT-qPCR analysis. Why are the three candidate genes of LAC gene family. This needs to be explained in detail and analyzed for reasons.
Response: We selected the three laccase genes for RT-PCR validation because they are meaningful in this study that they may have roles in pericarp browning in red rice, based on our bioinformatics analysis. The three genes were in 56 DEGs specifically upregulated in RP28, and their presence was highlighted in gene ontology analysis. Several studies have been reported that laccase has a function to be related to the seed coat browning by the oxidation of phenolic compounds in various plant species, and we described it by taking one paragraph in the Discussion section (lines 404-420).
Reviewer 2 Report
The manuscript entitled “Comparative biochemical and transcriptome analyses reveal potential candidate genes related to pericarp browning in red rice” and authored by Lee and colleagues, deals with the investigation of proanthocyanidins biosynthesis and role during the maturation of red rice.
The manuscript contains information that can seriously contribute to knowledge in this field. It appears well written and structured, although several typos are present in the main text. However, I do not feel that this would be a problem that would compromise its publication in Agriculture.
However, some revisions need to be made before I can consider this manuscript suitable for publication in the journal. Below is a series of comments, listed point by point:
KEYWORDS: The keywords should be completely changed. The utility of these terms is to facilitate the search of the article using common scientific search engines (PubMed, GoogleScholar, Scopus, etc.), which rely on the terms contained in title, abstract, and keywords. Consequently, using terms that are already in these sections as keywords is inappropriate. I strongly suggest that the keywords be changed before re-submission and add new ones (max 10).
INTRODUCTION: this section is really well written. The authors describe what is currently known about PAC biosynthesis, however, some important aspects should better emerge in their description. In particular, (i) the importance of PACs at the nutraceutical level, as these compounds have been shown to be actively involved in several human physiological processes, such as lowering cholesterol and blood glucose (REF), as well as in antiadhesive mechanisms against viruses (REF) and bacteria (REF); (ii) the distribution of PACs in the plant kingdom is quite limited, and mainly divided into red-colored plants (anthocyanins) that have PACs, and non-red-colored plants that have PACs. This different distribution is given by the fact that the genetic makeup of plants either express or do not express certain enzymes involved in the conversion of leucoanthocyanins (uncolored) to anthocyanins (red) and then to PACs (uncolored), or directly convert leucoanthocyanins (uncolored) to PACs (uncolored). These enzymes are ANS, ANR, and LAR; (iii) although the synthesis of individual flavan-3-ols is known, several studies have been performed with the aim to identify and describe the mechanism related to the transport of PAC precursors from the RE cytosolic face to plant vacuole, but until now, a precise transport mechanism of individual flavan-3-ol monomers has not been well identified; (iv) finally, it is also not well understood how different monomers can polymerize into PACs. The authors should implement this part. The necessary information can be found in this recent review: 10.3390/antiox10081229.
MATERIALS AND METHODS:
(i) Why did the authors not perform the DMAc assay to quantify total PACs? this is a widely and universally used method to quantify total PAC content, as well as being the most precise and accurate.
(ii) The authors correctly performed statistical analyses on the biochemical quantification data; however, a paragraph describing the analyses used is not present. Please add it.
RESULTS:
(i) Panel E in Figure 1 is dimly visible. The authors should look for a way to make the data understandable.
(ii) Figure 6 is also scarcely visible. The authors could divide the panels into several figures, or present one figure that takes up the whole page.
DISCUSSION:
(i) The authors mentioned several times in the text that PACs are pigments. However, these compounds are colorless. They become colored (green or red) as a result of chemical complexation reaction or acid hydrolysis, which are normally used to quantify their content within plant matrices. Please correct.
CONCLUSION: The concluding section should be better argued, including highlighting potential future prospects.
Author Response
Reviewer 2
The manuscript entitled “Comparative biochemical and transcriptome analyses reveal potential candidate genes related to pericarp browning in red rice” and authored by Lee and colleagues, deals with the investigation of proanthocyanidins biosynthesis and role during the maturation of red rice.
The manuscript contains information that can seriously contribute to knowledge in this field. It appears well written and structured, although several typos are present in the main text. However, I do not feel that this would be a problem that would compromise its publication in Agriculture.
Response: We appreciate the reviewer’s time and effort to review our manuscript.
However, some revisions need to be made before I can consider this manuscript suitable for publication in the journal. Below is a series of comments, listed point by point:
KEYWORDS: The keywords should be completely changed. The utility of these terms is to facilitate the search of the article using common scientific search engines (PubMed, GoogleScholar, Scopus, etc.), which rely on the terms contained in title, abstract, and keywords. Consequently, using terms that are already in these sections as keywords is inappropriate. I strongly suggest that the keywords be changed before re-submission and add new ones (max 10).
Response: We have revised the keywords according to the reviewer’s suggestion (Line 31-32).
INTRODUCTION: this section is really well written. The authors describe what is currently known about PAC biosynthesis, however, some important aspects should better emerge in their description. In particular, (i) the importance of PACs at the nutraceutical level, as these compounds have been shown to be actively involved in several human physiological processes, such as lowering cholesterol and blood glucose (REF), as well as in antiadhesive mechanisms against viruses (REF) and bacteria (REF); (ii) the distribution of PACs in the plant kingdom is quite limited, and mainly divided into red-colored plants (anthocyanins) that have PACs, and non-red-colored plants that have PACs. This different distribution is given by the fact that the genetic makeup of plants either express or do not express certain enzymes involved in the conversion of leucoanthocyanins (uncolored) to anthocyanins (red) and then to PACs (uncolored), or directly convert leucoanthocyanins (uncolored) to PACs (uncolored). These enzymes are ANS, ANR, and LAR; (iii) although the synthesis of individual flavan-3-ols is known, several studies have been performed with the aim to identify and describe the mechanism related to the transport of PAC precursors from the RE cytosolic face to plant vacuole, but until now, a precise transport mechanism of individual flavan-3-ol monomers has not been well identified; (iv) finally, it is also not well understood how different monomers can polymerize into PACs. The authors should implement this part. The necessary information can be found in this recent review: 10.3390/antiox10081229.
Response: (i) We have added the beneficial effects of PAs in human health (lines 45-46). (ii) We have revised the sentence according to the reviewer’s suggestion (lines 56-59). (iii and iv) We think the reviewer made good suggestion. In our study, unfortunately, we could not obtain meaningful results related to transport and polymerization of PAs such as candidate genes, so we did not describe them in detail. Nonetheless, if the reviewer still wants to elaborate on transport and polymerization, we will do it.
MATERIALS AND METHODS:
(i) Why did the authors not perform the DMAc assay to quantify total PACs? this is a widely and universally used method to quantify total PAC content, as well as being the most precise and accurate.
Response: We fully agree that the DMAc assay is a widely and universally used method to measure the amount of PAs. The DMACA staining assay, which works on the similar principle as the DMAc assay is also one of the experimental methods that intuitively show whether PAs are accumulated or not. We focused on the timing of the accumulation of PAs rather than measuring the exact amount of PAs, therefore we adopted the DMACA staining method which has the advantage of experimental simplicity.
(ii) The authors correctly performed statistical analyses on the biochemical quantification data; however, a paragraph describing the analyses used is not present. Please add it.
Response: We have added a paragraph for statistical analysis (lines 163-165).
RESULTS:
(i) Panel E in Figure 1 is dimly visible. The authors should look for a way to make the data understandable.
Response: We have prepared high resolution images.
(ii) Figure 6 is also scarcely visible. The authors could divide the panels into several figures, or present one figure that takes up the whole page.
Response: We have prepared high resolution images and we think they will be used for publication. Additionally, we found typos and errors in Figure 6 and have revised them.
DISCUSSION:
(i) The authors mentioned several times in the text that PACs are pigments. However, these compounds are colorless. They become colored (green or red) as a result of chemical complexation reaction or acid hydrolysis, which are normally used to quantify their content within plant matrices. Please correct.
Response: We have corrected the manuscript according to the reviewer’s comment.
CONCLUSION: The concluding section should be better argued, including highlighting potential future prospects.
Response: We have revised the manuscript according to the reviewer’s comment.
Reviewer 3 Report
In this manuscript, the authors compared the transcriptomes of immature seeds (14 DAF and 28DAF) between a red rice variety RBRC_OS_RP1 and a white rice variety Dongan, and the differentially expressed genes (DEGs) were then identified and analyzed. Finally, the gene candidates potentially responsible for pericarp browning were selected from the DEGs based on in silico analysis. The only validation experiment is qRT-PCR analysis. Because only one red rice variety and one white rice variety used for transcriptomic analysis, I believe that many DEGs may be variety specific rather than the DEGs between red rice and white rice. Therefore, I think that additional functional research is needed to validate the functions of these candidate genes for pericarp browning.
Author Response
Reviewer 3
In this manuscript, the authors compared the transcriptomes of immature seeds (14 DAF and 28DAF) between a red rice variety RBRC_OS_RP1 and a white rice variety Dongan, and the differentially expressed genes (DEGs) were then identified and analyzed. Finally, the gene candidates potentially responsible for pericarp browning were selected from the DEGs based on in silico analysis. The only validation experiment is qRT-PCR analysis. Because only one red rice variety and one white rice variety used for transcriptomic analysis, I believe that many DEGs may be variety specific rather than the DEGs between red rice and white rice. Therefore, I think that additional functional research is needed to validate the functions of these candidate genes for pericarp browning.
Response: We appreciate the reviewer’s time and effort to review our manuscript. We agree with the reviewer’s comment that many DEGs may be variety specific rather than the DEGs between red rice and white rice. Therefore, our study focused more on previously known biosynthetic pathways and structural genes. We also agree with the reviewer’s comment that additional functional research is needed to validate the functions of these candidate genes for pericarp browning. Unfortunately, at this point, we do not have various red pericarp rice resources, and it will take at least four months from seeding to getting grain samples at each developmental stage. We think that the reviewer’s suggestion could become the subject of further research.
Round 2
Reviewer 3 Report
No comments